# Roles of E3 Ubiquitin Ligases in Plant Responses to Abiotic Stresses

**DOI:** 10.3390/ijms23042308

**Published:** 2022-02-19

**Authors:** Shuang Wang, Xiaoyan Lv, Jialin Zhang, Daniel Chen, Sixue Chen, Guoquan Fan, Chunquan Ma, Yuguang Wang

**Affiliations:** 1Engineering Research Center of Agricultural Microbiology Technology, Ministry of Education, Heilongjiang University, Harbin 150080, China; suangsuang0923@163.com (S.W.); zx8262889@163.com (J.Z.); 2School of Life Science and Technology, Harbin Institute of Technology, Harbin 150080, China; qq1038819638@gamil.com; 3Judy Genshaft Honors College and College of Arts and Sciences, University of South Florida, Tampa, FL 33620, USA; chend@usf.edu; 4Plant Molecular and Cellular Biology Program, Department of Biology, Genetics Institude, University of Florida, Gainesville, FL 32610, USA; schen@ufl.edu; 5Industrial Crops Institute, Heilongjiang Academy of Agricultural Sciences, Harbin 150086, China; fgq_520@126.com

**Keywords:** ubiquitination, ubiquitin ligase, salt stress, drought stress, temperature stress, abscisic acid

## Abstract

Plants are frequently exposed to a variety of abiotic stresses, such as those caused by salt, drought, cold, and heat. All of these stressors can induce changes in the proteoforms, which make up the proteome of an organism. Of the many different proteoforms, protein ubiquitination has attracted a lot of attention because it is widely involved in the process of protein degradation; thus regulates many plants molecular processes, such as hormone signal transduction, to resist external stresses. Ubiquitin ligases are crucial in substrate recognition during this ubiquitin modification process. In this review, the molecular mechanisms of plant responses to abiotic stresses from the perspective of ubiquitin ligases have been described. This information is critical for a better understanding of plant molecular responses to abiotic stresses.

## 1. Ubiquitination Modification

Proteins are not only structural molecules but also action molecules in all life forms. The lifespans of protein molecules in a cell range from less than a minute to many days [1]; thus, protein production and degradation play a fundamental role in all cells during plant growth, development, and responses to environmental changes. By regulating the abundance of key proteins plants are able to modulate signaling events, ensuring that the proper response is initiated when required and only for the appropriate length of time [2]. Protein degradation requires the degradation of signals—*N*-degrons/*C*-degrons—including not only adjacent sequence motifs but also internal lysine residues modified by polyubiquitin. All 20 amino acids function as destabilizing *N*-terminal residues, which complicates the *N*-degron pathways [1].

The endoplasmic reticulum-associated protein degradation (ERAD) pathway [3,4], ubiquitin proteasome system (UPS) that degrades ubiquitinated proteins via 26S proteasome [5], and the lysosome-mediated intracellular degradation pathway [6] are all related to ubiquitination modification. Ubiquitination is a multistep process governed by ubiquitin-activating enzymes (E1s), ubiquitin-conjugating enzymes (E2s), and ubiquitin-ligase enzymes (E3s) that successively ligate ubiquitin to substrate proteins [7]. Due to conservation, the constantly high expression levels of ubiquitin in different tissues and organs allow it to be utilized as a housekeeping gene marker for gene expression analysis in plants [8]. This also highlights the prevalence of ubiquitinated proteins in the proteome as a major proteoform [9,10].

A single ubiquitin can be attached to one (monoubiquitination) or multiple (multimonoubiquitination) lysine residues within the substrate protein. Alternatively, repetition of the conjugation process can generate a polyubiquitin chain on a single lysine of the substrate (polyubiquitination) [2]. After the enrichment of ubiquitinated peptides [11,12,13], the lysine sites modified by ubiquitin can be identified by mass spectrometry and immunological methods [14,15]. In addition, mass spectrometry has been used to achieve the quantification of ubiquitin levels [16,17]. However, the quantitative study of ubiquitin in plant cells has not been reported. Different ubiquitin pools were quantitatively analyzed in HEK293 and MEF cell lines, mouse brains, and human frontal cortexes through differential affinity chromatography combined with protein standard absolute quantification (PSAQ) mass spectrometry technology (Ub-PSAQ). The results showed that the proportional distribution of ubiquitin, ‘free’ ubiquitin, ubiquitin chains, and monoubiquitinated-modified conjugates were different in different cell types [15]. In addition, under the treatment of proteasome inhibitor MG-132, the proportions of the ubiquitin pools changed [12]. Studies in different model cells have shown that the redistribution mechanism of ubiquitin pools may be different, including de novo synthesis of ubiquitin and the transformation of multi-ubiquitin chains to mono-ubiquitin chains [15].

Ubiquitin has seven lysine residues (K6, K11, K27, K29, K33, K48, and K63) that provide sites for the formation of different isopeptide chain linkages. In addition, the free amino group of the *N*-terminal methionine (Met1) of ubiquitin can also be modified by other free ubiquitin molecules in tandem, which further increases the diversity and complexity of ubiquitin chains. A polyubiquitin chain can be homogeneous when the same lysine residue is used to build the polymer, or of mixed topology when different lysine residues are used to create the ubiquitin–ubiquitin linkage [18]. The Lys48-linked chains are the predominant linkage type in cells [19]. Pproteins modified by K48 will be degraded; however, Lys63 chains are non-degradable modification chains. Additionally, there are some atypical ubiquitin modifications on different sites of amino acid residues (Met1, Lys6, Lys11, Lys27, Lys29, and Lys33) [20]. Ubiquitin is also modified by small chemically-distinct post-translational modifications (PTMs), such as phosphorylation and acetylation. Six out of the seven lysine residues in ubiquitin can become acetylated [21]. There are multiple sites on ubiquitin that can be phosphorylated, such as Thr7, Thr12, Thr14, Ser20, Thr22, Thr55, Thr67, Tyr59, and Ser59. Moreover, multiple ubiquitin lysine residues (Lys6, Lys11, Lys27, Lys48, and Lys63) can be targeted for SUMOylation [22]. Compared to other PTMs, ubiquitination is a relatively complicated modification in the cells.

## 2. Classification and Functions of E3 Ubiquitin Ligases

Plants engage various regulations at the levels of transcription, translation, and post-translational modifications to mediate stress perception, signaling, and responses [2]. PTMs are at the heart of many cellular signaling events [23]. Protein ubiquitination is one of the most prevalent PTMs as it regulates a plethora of cellular processes in distinct manners [24] and orchestrates a spectrum of different cellular processes, including substrate degradation, protein localization, and enzyme activation or inactivation. Ubiquitin ligases regulate protein abundance to ensure that the stress response is initiated only when required, maintained at an appropriate intensity and eliminated once it is no longer needed [25].

Ubiquitination begins with the activation of ubiquitin by E1 followed by the transfer of ubiquitin to E2, whose active-site cysteine forms a thioester bond with the *C*-terminal carboxyl group of ubiquitin. Substrate-recruiting E3 interacts with the E2–ubiquitin (E2–Ub) intermediate, allowing for the transfer of ubiquitin to the target [18,26]. The specific combination of E2 and E3 enzymes recruited to a substrate dictates the chain linkage type [27]. At the end of the three-enzyme cascade, E3 exhibits strict control over both the efficiency and substrate specificity of the ubiquitination reaction [26].

Ubiquitin ligase E3 has varying isoforms, which can be divided into the following categories according to its different catalytic domains: RING (really interesting new gene), HECT (homology to E6-associated carboxyl terminus), and U-box [2,18]. At the same time, HECT, U-box, and RING all belong to a single subunit of E3, while the multi-subunit refers to an enzyme with F-box sequence characteristics [28]. RING E3 is one of the key types of ubiquitin ligase. The RING-finger domain exists not only in the single subunit RING ubiquitin ligase and RBR ubiquitin ligase (ring between ring) [29,30] but also in the multi-subunit cullin-RING E3 [31]. The U-box domain is similar to the RING domain in that it is essentially a modified RING-finger domain [32]. The U-box domain is a domain of about 70 amino acid residues in both lower and higher organisms [33]. The corresponding proline residue of *S. cerevisiae* UFD2 proline 924 is completely conserved in the U-box proteins of mammals and other organisms [34]. Hatakeyama, S. et al. demonstrated that conserved proline is essential for the function of U-box proteins because mutation of the U-box domain (P1140A) abolishes the E3 activity of UFD2a [34]. Therefore, U-box ligase E3 is considered as the third ubiquitin ligase E3 isoform besides RING and HECT. As with RING ligases, U-box ligases also do not form E3–Ub intermediates [33]. There are 2, 21, and 77 U-box proteins in yeast, humans, and rice, respectively [34,35,36]. An increasing number of U-box proteins indicates not only their importance in governing cellular processes that are specific to plants but also the wide range of functional involvement that these proteins could have as part of regulated plant growth and development [32]. RBR is a single subunit ubiquitin ligase containing two RING domains, one of which contains cysteine residues, which can bind to ubiquitin to form an E3–Ub intermediate [37]. 

As a key enzyme in ubiquitination modification, ubiquitin ligase plays a role in determining substrate specificity. E3 can recognize substrates through specific sequences, such as the APC/C complex recognition D-box motif [38], and the KEN-box and HECT family recognition PY motifs of substrates [39]. E3 can also recognize substrates through adaptors. For example, the NEDD4 family ubiquitin ligase SMURF1 uses the adaptor Smad7 to bind TGFβ and mediate the ubiquitination degradation of TGFβ [40].

## 3. Ubiquitin Ligases in Plant Abiotic Stresses

As sessile organisms, plants have evolved a variety of complex adaptive mechanisms to cope with adverse environmental conditions, including the maintenance of ion homeostasis, accumulation of antioxidant enzymes, and synthesis of compatible products [41,42,43]. Furthermore, stress-related cis-acting elements or transcription factors (TFs) as well as stress response genes can be activated in plant stress responses and adaptations. Abiotic stressors, such as soil salinity, drought, or extreme temperature variations, impair crop productivity and are therefore the main causes of reduced crop yield [44,45].

### 3.1. Salt Stress 

Human population increase, dysfunctional drainage, and irrigation-aggravated soil salinization [46] cause there to be a high concentration of Na^+^ in the soil, leading to saline–alkaline land. This hyperosmotic condition hinders the absorption of water by plants and nutrients in the soil, which results in a significant decline in crop productivity [47,48]. Salt stress has both osmotic and ionic or ion-toxicity effects on cells [49]. Proteins critical for salt-stress signaling, ion and water transport, redox homeostasis, and metabolism are regulated to bring about ionic and water homeostasis and cellular stability under salt stress [49]. 

#### 3.1.1. E3s Participate in the SOS Pathway and MAPK Cascade 

The salt overly sensitive (SOS) pathway is a well known signaling module that controls cellular ion homoeostasis (Figure 1) [50,51]. During salt stress, extracellular Na^+^ induces a transient rise in the cytosolic Ca^2+^ concentration, which triggers the SOS pathway [52]. A calcium-derived signal activates the SOS pathway by binding to the SOS3 and ScaPB8/CBL10 calcium-binding proteins, which activate the SOS2 protein kinase to regulate the SOS1, a plasma membrane (PM) Na^+^/H^+^ antiporter. SOS2 is a key regulator in the SOS pathway, relaying the signal downstream through changes in protein phosphorylation [52,53]. Interestingly, a flowering time regulator, GIGANTEA (GI) can prevent SOS2 from phosphorylating SOS1 in the absence of salt stress [54]. This mechanism may allow plants to maintain regular growth and development programs under adverse circumstances. In addition to the SOS pathway, the mitogen-activated protein kinase (MAPK) cascade functions during biotic and abiotic stress responses by receiving extracellular signals and activating the expression of downstream target genes [55]. A MAPK cascade is minimally composed of distinct combinations of at least three protein kinases: a MAPKKK (MAP3K/ MEKK/MKKK), a MAPKK (MKK/MEK), and a MAPK (MPK), which form a cascade of activation via transphosphorylation in a sequential manner [56,57]. 

The SOS and MAPK pathways are closely regulated by protein ubiquitination. *EST1* (Ever shorter Telomeres 1) encodes an F-box protein, which, as a subunit of SCF E3 ubiquitin ligase, negatively regulates Arabidopsis in response to salt stress [58]. The salt-tolerant phenotype of the *est1* mutant is also dependent on the function of SOS1, which is downstream of *EST1*. However, there is no direct interaction between SOS1 and *EST1*. On the contrary, *EST1* directly interacts with MKK4 and negatively regulates its protein level, which may lead to decreased activity of the MKK4-MPK6 cascade reaction, resulting in decreased activity of the Na^+^/H^+^ antiporter. Compared with the wild type (WT), the *est1* mutant showed higher PM Na^+^/H^+^ antiporter activity, resulting in lower intracellular Na^+^ concentration, which led to lower Na^+^ accumulation during salt stress, and thus higher salt tolerance [58]. Another example was E3 ubiquitin ligase *IbATL38* (*Ipomoea batatas* Arabidopsis Toxicos en Levadura 38), which is localized to the nucleus and cell membrane. The expression pattern of *IbATL38* was different under different stress treatments and the expression was induced under salt and ABA treatment. Under salt stress, *A. thaliana* plants overexpressing *IbATL38* showed a trend of improved growth and increased expression of stress response genes such as *AtSOS1*, *AtSOS2*, *AtSOS3*, *AtRD29A*, and *AtKIN2*. The results suggested that *IbATL38* was involved in plant responses to salt stress as a positive regulatory factor [59].

#### 3.1.2. E3s Participate in the ABA Signaling Pathway 

Under normal growth conditions, the E3 ubiquitin ligase RGLG (RING domain ligase) is localized in the PM [60]. The myristoyltransferase NMT1 is decreased under ABA or salt stress conditions. This in turn inhibits the myristoylation modification of RGLG, resulting in the migration of RGLG to the nucleus. This change of subcellular localization enables its interaction with PP2C in the nucleus, leading to degradation of ubiquitinated PP2C and thus turning on the ABA pathway [61] (Figure 1).

#### 3.1.3. E3s Participate in the Flowering Pathway

In plants, 14-3-3 proteins are recognized as mediators of signal transduction and function in both plant development and stress response [62]. A foxtail millet 14-3-3 protein *SiGRF1* (*Setaria italica* growtn-regulating factor 1) is involved in flowering under salt stress and ubiquitinated by E3 ubiquitin ligase SiRNF1/2 (*Setaria italica* RING finger protein 1/2). The *SiGRF1* gene may regulate the initiation date of flowering in plants exposed to salt stress by up-regulating the transcription level of *WRKY71* to promote *Flowering Locus T* (*FT*) and *LEAFY* (*LFY*) expression to act against the inhibition by DELLAs, etc. Additionally, *SiGRF1* helps to avoid salt stress by accelerating plant flowering time [63].

#### 3.1.4. E3s Participate in ROS Homeostasis

A high concentration of reactive oxygen species (ROS) causes a direct impact upon biological membranes, disrupts macromolecules, promotes cell senescence, and induces irreversible cellular damage [64]. Two eminent types of defensive responses are enzymatic response and non-enzymatic response [65]. In Arabidopsis, protein arginine methyltransferase 4b (PRMT4b) can methylate histones on the chromatin of an *A. thaliana* ascorbate peroxidase 1 (AtAPX1) gene and a glutathione peroxidase 1 (AtGPX1) gene, consequently increasing their expression and leading to enhanced stress resistance [66]. On the other hand, PRMT4b interacts with the E3 ubiquitin ligase paraquat tolerance 3 (PQT3), resulting in the degradation of PRMT4b through the 26S proteasome. Therefore, PQT3 is regarded as a negative regulator through indirect regulation of antioxidant enzymes [67]. *OsPQT3* is a homologous gene of *AtPQT3*, which also plays a negative regulatory role in plant resistance to paraquat and salt stress. Compared with the WT, the *Ospqt3* mutant had higher APX, GPX, and SOD activity, as well as less ROS accumulation [64]. The study on the salt tolerance of rice seedlings and vegetative growth showed that the survival rate, seed setting rate, and tillering rate of the mutant were higher than those of the WT after salt stress treatment, indicating that the mutant was more resistant to salt stress [68].

#### 3.1.5. E3s Participate in the ERAD Pathway

Previous studies in *A. thaliana* have found that the E3 ubiquitin ligase MfSTMIR (*Medicago falcata* salt tunicamycin-induced RING finger protein), an endoplasmic reticulum (ER) protein, ubiquitinates MfCPY*, which contains the G-to-R mutation at residue 255 [69]. MfSTMIR can interact with the E2 MtUBC32 (ubiquitin-conjugating enzyme 32) and Sec61-translocon subunit MtSec61c that can bind proteasome 19S regulatory particle, which can extract an ERAD substrate from the ER. However, MfSTMIR does not degrade Sec61c as a ubiquitination substrate, which helps reduce stress on the ER while under salt stress. Therefore, MfSTMIR can act as a positive regulator in response to salt stress [70].

### 3.2. Drought Stress

#### 3.2.1. E3s Participate in DREB2A-Mediated Stress Signaling

Drought and salt have overlapping signals and both result in hyperosmotic stress and accumulation of phytohormone abscisic acid (ABA) [41] (Figure 1 and Figure 2). Dehydration-responsive element-binding protein 2A (DREB2A) is a key transcriptional activator that induces transcription of many drought-response genes [71]. In wheat and Arabidopsis, DREB2A interacting protein 1 (DRIP1) and DRIP2 ubiquitinate DREB2A, resulting in its degradation by the 26S proteasome, which keeps it at a very low level. Under drought stress, TaSAP5 (*Triticum aestivum* stress-associated protein) acts as an E3 ubiquitin ligase to mediate DRIP [72] and HSP90C (chloroplast heat shock protein 90) ubiquitination [73] and degradation. Thereby, ubiquitination of DREB2A by DRIPs is decreased, leading to sufficient accumulation of DREB2A to initiate the expression of downstream genes. Thus, it can be concluded that SAP5 plays a role as a positive regulator in plant drought responses (Figure 2).

#### 3.2.2. E3s Participate in MAPK Cascades

Another example of the interplay between drought and salt stress is E3 ubiquitin ligase RGLG (Figure 1 and Figure 2). Under drought stress, the modification of RGLG was also affected, resulting in a shift from the PM to the nucleus. It was found that RGLG interacts with a drought-inducible TF ethylene response factor53 (AtERF53) and mediates its degradation [60]. Meanwhile, RGLG can also interact with ubiquitin conjugating enzyme UBC13 to mediate ubiquitination of PM-located auxin carrier protein pin-formed 2 (PIN2) to control its turnover, thus affecting auxin transport. Mutations in RGLG1/2 caused an arrest of endocytosis of PIN2 and decreased auxin levels, eventually leading to the branching of root hairs [74,75]. Further studies demonstrated that RGLG could respond to drought stress by participating in the MAPK pathway, and RGLG ubiquitinated MAPKKK18 to promote its degradation, thus playing a negative regulatory role in plant drought tolerance [76].

#### 3.2.3. E3s Participate in the ABA Signaling Pathway

*SpRing* is a RING-H2 ubiquitin ligase found in tomatoes and is specifically located in the ER. ABA, drought, and salt stress could induce the expression of *SpRing*. Under salt stress, the chlorophyll content of the *SpRing*-silenced plants decreased and produced more malondialdehyde and H_2_O_2_. In *SpRing* overexpression Arabidopsis plants, *NCED3* (*Nine-Cis-Epoxycarotenoid Dioxygenase 3*), *RD29A* (*Responsive to Deciccation29A*), and *Rab18* exhibited increased expression levels [77]. In Arabidopsis, RING ubiquitin ligase AtAIRP1 (*A. thaliana* ABA-insensitive RING 1) and AtAIRP2 are positive regulators in ABA-dependent responses to drought stress. AtAIRP1 is rapidly induced by drought and ABA. It positively regulates ABA-promoted stomatal closure, which may reduce transpirational water loss in response to dehydration stress [78,79]. AtAIRP2 down-regulates ATP1/SDIRIP1 (AtAIRP2 target protein 1/ SDIR1-interacting protein 1) through UPS during Arabidopsis seed germination [80]. *AtAIRP3/LOG2* (loss of glutamine dumper 2) was up-regulated by high salinity, drought, and ABA treatments. Under ABA treatment, the mean stomatal diameter of leaves of both WT and the *atairp3/log2* mutant decreased, but the extent of the reduction in the mutant was not as great as that in the WT. RD21 (responsive to desiccation 21) was initially isolated as a drought-induced cysteine proteinase [81]. The RD21 protein was ubiquitinated in vitro by *AtAIRP3/LOG2* and degraded by 26S proteasome. Thus, *AtAIRP3/LOG2* is a positive regulator of the ABA-mediated drought and salt stress tolerance [82]. *AtAIRP4* acts as a positive regulator of ABA-mediated drought avoidance and a negative regulator of salt tolerance. The *atairp4* mutant showed reduced sensitivity of root elongation and stomatal closure to ABA, whereas plants with overexpressed *AtAIRP4* were hypersensitive to salt and osmotic stresses during seed germination. The transcriptional abundances of ABA-responsive genes in overexpression plants were higher than those in the WT and *atairp4* plants [83]. 

PEG, ABA, and NaCl treatments significantly induced the expression of *TaDIS1* (*T. aestivum* drought-induced SINA protein 1) in wheat. Overexpression of *TaDIS1* in Arabidopsis reduced the tolerance to drought stress and increased sensitivity to ABA during seed germination [84]. *TaDIS1* interacts with TaSTP (*T. aestivum* salt tolerant protein) in the Golgi apparatus and degrades TaSTP via the 26S proteasome pathway [85]. A cytoplasmic *PnSAG1* is a PUB-ARM ubiquitin E3 ligase from the Antarctic moss *Pohlia nutans.* Its transcription was rapidly induced by ABA, salt, and drought stress. Overexpression of *PnSAG1* in *A. thaliana* increased the sensitivity to salt and ABA during seed germination and decreased the expression levels of salt-/ABA-related genes [86]. In Arabidopsis, a C3H2C3 RING-type E3 ligase *ATL61* plays an important role in drought tolerance. Point mutation of *ATL61*^H109A, H122A^ (*mATL61*) abolished E3 ubiquitin ligase activity, and *mATL61* overexpression lines exhibited similar ABA-related phenotypes as the WT plants. *ATL61*-overexpression plants exhibited ABA hypersensitivity and were more tolerant to drought, while the *atl61* mutant plants were insensitive to ABA [87]. 

#### 3.2.4. E3s Participate in Phosphorylation and Ubiquitination Crosstalk 

Another RING-H2 E3 ubiquitin ligase, CHY zinc-finger and RING protein1 (CHYR1), was induced by drought stress and ABA treatment. It is located in the cytoplasm and nucleus [88]. CHYR1 positively regulates ABA-induced ROS production and stomatal closure [88]. In addition, studies have found that CHYR1 is involved in plant resistance to pathogen attack [81]. One mechanism is that CHYR1 mediates the turnover of *WRKY70*, which is known to be involved in plant osmotic stress and immune responses. In normal growth conditions, *WRKY70* is mainly nonphosphorylated, so few phosphorylated WRKY70s will be degraded by the CHYR1-mediated 26S proteasome pathway. Upon pathogen attack, *WRKY70* mainly becomes phosphorylated at Thr22 and Ser34 residues. The phosphorylated *WRKY70* (*WRKY70*-P) is recognized by CHYR1 and gets ubiquitinated and degraded [89]. During plant immunity, the TF *SARD1* (SAR deficient 1) and CBF60g (calmodulin-binding Factor 60-like g) mediate the synthesis of the plant defense hormone salicylic acid (SA) by regulating the transcription of isochorismate synthase 1 (ICS1) [90]. Overexpression of *WRKY70* activates the expression of SA-responsive pathogenesis-related (*PR*) genes, whereas silencing of *WRKY70* leads to up-regulated expression of jasmonic acid (JA)-responsive genes. *WRKY70* can positively regulate the expression of *SNC2* (suppressor of NPR1, constitutive 2) and downstream genes *SARD1* and *CBP60g*. In turn *SARD1* and *CBP60g* can also bind to the promoter of *WRKY70*, forming a circular structure that promotes the expression of each gene [91,92,93]. In general, this is a great example of phosphorylation and ubiquitination crosstalk in plant responses to drought and biotic stress.

### 3.3. Temperature and Cold Stress 

Cold stress causes substantial loss in global agricultural productivity [94,95]. Under low temperatures, plants exhibit a variety of cold-induced physiological and biochemical responses, such as production of ROS, changes in membrane lipid composition, and changes in osmolytes [96,97,98]. Expression of C-repeat binding factor (*CBF*) genes is rapidly induced by cold, and their translational products directly bind to the promoters of cold-regulated genes. This activates their expression and thus protects plants from damage caused by cold stress (Figure 3) [96,99]. 

#### 3.3.1. E3s Participate in the Cold Signaling Pathway 

Cold stress is sensed via membrane proteins, such as a cold receptor COLD1 (chilling-tolerance divergence 1) [100], leading to a cytosolic Ca^2+^ spike. CPKs (calcium-dependent protein kinases) and CBLs (calcineurin B-like proteins)-CIPKs (CBL-interacting protein kinases) may mediate the Ca^2+^ signal to activate an MAP kinase cascade [49,101,102]. MPK3 and MPK6 mediate the phosphorylation and destabilization of a TF ICE1 [103,104], and MPK6 phosphorylates MYB15 and reduces its transcriptional activation of the *CBF3* gene [105]. ICE1 (*CBF* expression 1) interacts with MYB15 and also directly binds to the *CBF3* promoter to negatively regulate its expression, thereby modulating plant freezing tolerance [106,107].

Under cold stress, E3 ubiquitin ligases PUB25 and PUB26 can directly mediate the ubiquitination of MYB15 and lead to its degradation [108]. In addition, open stomata 1 (OST1)/SnRK2.6 is activated by cold stress and phosphorylates PUB25 and PUB26. Phosphorylation of PUB25 and PUB26 enhances MYB15 degradation in plants [105]. Therefore, PUB25 and PUB26 are located upstream of the MYB15 and downstream of OST1 to positively regulate *CBF* expression and plant cold responses [108,109]. 

#### 3.3.2. E3s Participate in Phosphorylation and Ubiquitination Crosstalk 

*N*-myristoylation is not only important for protein—membrane interactions but it is also important for protein—protein interactions [110,111]. Under normal conditions, a PM-localized EGR2 (clade-E growth-regulating 2) phosphatase is modified by NMT1 and interacts with OST1 to inhibit OST1 activity. At low temperatures, the interaction between EGR2 and NMT1 is weakened, resulting in the inhibition of the myristic acylation of EGR2, which partially contributes to increased OST1 activity in response to low-temperature stress [112].

#### 3.3.3. E3s Participate in the Flowering Pathway

Light and temperature are two important environmental determinants of flowering time. Constans (CO) is a central activator of photoperiodic flowering [113]. CO protein is stable during light but is degraded rapidly in darkness. Dark-induced CO protein degradation is mediated by a ubiquitin—proteasome system, with phyB signaling through HOS1 (high expression of osmotically responsive gene 1), promoting CO degradation in the morning under long daylight conditions. At night, blue and far-red signals inhibit the activity of COP1 (constitutive photomorphogenic 1), leading to an increase in CO abundance. In darkness, the CO protein is rapidly degraded through COP1-mediated ubiquitination system. At low temperatures, cold-activated HOS1 induces CO degradation, resulting in delayed flowering. In summary, CO acts as a molecular hub that integrates light and cold stress signals into photoperiodic flowering [114,115].

### 3.4. Temperature and Heat Stress

Heat stress affects plant seed germination, photosynthesis, respiration, water transpiration, and membrane stability [116]. As the terminal components of high temperature signal transduction, heat shock TFs (HSFs) trigger the transcription of heat-responsive genes encoding heat shock proteins (HSPs) and other heat-protective proteins. These include molecules such as ROS scavengers, enzymes involved in the biosynthesis of protective metabolites and osmolytes, apoptotic regulators, and other TFs [117,118,119]. Under stress conditions, HSPs interact with key heat-responsive proteins (e.g., HSFs) to form complexes, preventing them from denaturation. HSFs are then transported into the nuclei and form active trimers for transcription activation (Figure 4) [120,121]. In addition, HSPs act as molecular chaperons that target ubiquitin-mediated degradation of misfolded or damaged proteins through autophagy and/or 26S proteasome systems [122,123,124]. 

Under heat shock, the misfolded proteins increased rapidly, and the transcription level of U-box E3 ubiquitin ligase gene *SlCHIP* (*Solanum lycopersicum* carboxyl terminus of the HSC70-interacting proteins) in tomatoes increased at the early stage (Figure 4). Under heat stress, the photosynthetic capacity of *Slchip*-silenced tomato plants decreased. It is well known that photosynthesis is very sensitive to heat stress due to damage to photosynthetic organs. The response of chloroplasts to heat stress is crucial to reduce the damage and improve survival rate under high temperatures [125], suggesting that CHIP plays an important role in protecting chloroplasts from heat stress. In addition, silenced tomato plants also showed higher electrolyte permeability. These results indicated that tomato *SlCHIP* plays a key role in heat stress responses, mostly by targeting degradation of misfolded proteins produced during heat stress [126].

RING-H2-type E3 ubiquitin ligases (OsHTAS) localized to the nucleus and the cytoplasm are highly expressed in mesophyll cells. OsHTAS can interact with APX and 26S proteasome, thereby effectively regulating the accumulation of H_2_O_2_ and functions in the leaves to enhance heat tolerance, e.g., through modulating H_2_O_2_-induced stomatal closure [127]. E3 ubiquitin ligases tend to recognize misfolded proteins with the assistance of HSPs [128], and the presence of chaperones in OsHTAS complexes is not clear. Additionally, the complete sets of OsHTAS substrates in the nucleus and the cytoplasm, and in stomatal guard cells, need to be characterized using proteomics tools in the future.

## 4. Conclusions

Plants are sessile organisms. Environmental stresses cause perturbation of cellular metabolism and oxidative damage to cellular structures and molecules. Cell signaling in response to salt, drought, cold, and the stress hormone ABA largely relies on the SnRK family of protein kinases in plants. This mainly involves the activation of SnRK2 kinases to mediate several rapid responses, including gene expression regulation, stomatal closure, protein PTMs, changes in metabolism related to stress resistance, and plant growth modulation [41,129,130,131]. MAPKs have also been implicated in ABA signaling [132] and cold responses [46,90,91,92]. Of the different stresses, there are common themes, e.g., ROS and redox regulation, biosynthesis of protective metabolites/osmolytes, kinase regulation, protein phosphorylation, and ubiquitination. For example, hormone-induced protein ubiquitination plays a crucial role in determining the half-life of key regulators in plant stress responses. Plants utilize the UPS system to alter intracellular protein abundance, which is vital in responding to and resisting the environmental stressors. Activation of stress signaling may also involve ubiquitin-dependent degradation of negative regulators and/or accumulation of positive regulators [2]. Clearly, changes in the expression of the ubiquitin enzymes can alter plant responses to abiotic stressors [2]. PTM crosstalks, especially with ubiquitination are clearly demonstrated in plant responses to salt, drought, and cold [86,95,96]. Although ubiquitin ligases in plant responses to salt, drought, cold, and heat stressors may involve different pathways (e.g., ABA signaling, MAPK cascade, and ROS homeostasis), their direct substrates under the different conditions and in single cell types (e.g., guard cells) are largely unknown. For example, how *SiGRF1* interacts with the aforementioned GI function is an interesting question to be addressed in the future. Future research on large-scale proteomic discovery of the E3 ligase substrates in a spatial and temporal manner, characterizing the novel targets, elucidating PTM crosstalks, and constructing molecular networks in plant stress responses will fill critical knowledge gaps and improve understanding of the molecular mechanisms underlying plant resistance to abiotic stressors. The improved knowledge will have translational potential in improving crop resilience through molecular breeding, and thereby contributing to global food security.

## Figures and Tables

**Figure 1 ijms-23-02308-f001:**
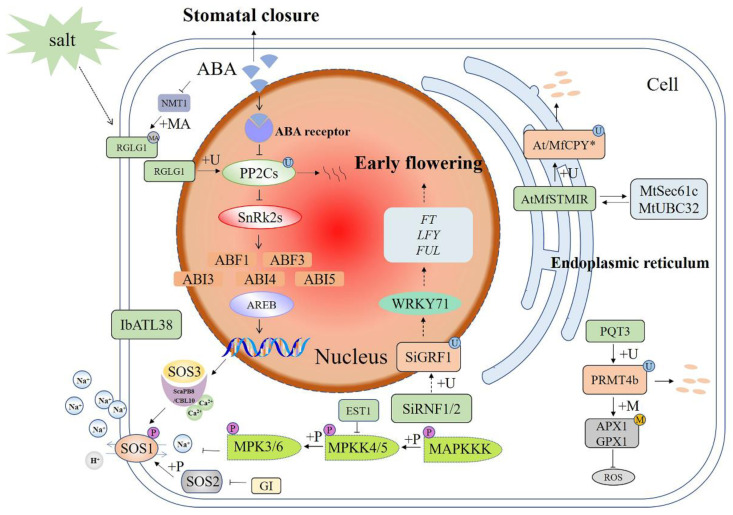
Schematic diagram of E3 ubiquitin ligases in different signaling pathways involved in plant responses to salt stress. The salt-stress pathways crosstalk with the ABA and MAPK pathways. E3 functions by ubiquitination of downstream target proteins. +P—phosphorylation; +U—ubiquitination; +MA—*N*-myristoylation; +M—methylation; ABA—abscisic acid; SOS—salt overly sensitive; GI—GIGANTEA; *IbATL38*—*Ipomoea batatas* Arabidopsis Toxicos en Levadura 38; *SiGRF1*—*Setaria italica* growtn-regulating factor1; *EST1*—Ever shorter Telomeres 1; SiRNF1/2—*Setaria italica* RING finger protein 1/2; RGLG—RING domain ligase; *FT*—*Flowering Locus T*; *LFY*—*LEAFY*; PRMT4b—protein arginine methyltransferase 4b; AtAPX1—*Arabidopsis thaliana* ascorbate peroxidase 1; GPX1—glutathione peroxidase 1; PQT3—paraquat tolerance 3; MfSTMIR—*Medicago falcata* salt tunicamycin-induced RING finger protein; MtUBC32—ubiquitin-conjugating enzyme 32; ROS—reactive oxygen species; MAPK—mitogen-activated protein kinase. A solid arrow shows a promoting effect or positive regulation; a dotted arrow shows that the specific mechanism of action is unclear; a horizontal line shows inhibition or negative regulation; a double arrow shows interacting proteins.

**Figure 2 ijms-23-02308-f002:**
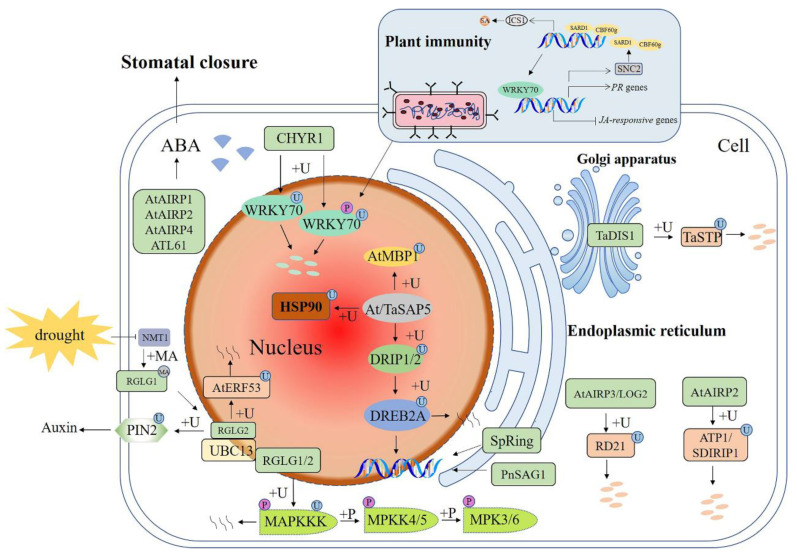
Schematic diagram of E3 ubiquitin ligases in signaling pathways involved in plant responses to drought stress. E3 functions by ubiquitination of downstream target proteins. +P—phosphorylation; +U—ubiquitination; +MA—*N*-myristoylation; +M—methylation; ABA—abscisic acid; DREB2A—dehydration-responsive element-binding protein 2A; DRIP1—DREB2A interacting protein 1; TaSAP5—*Triticum aestivum* stress-associated protein; AtAIRP1—*A. thaliana* ABA-insensitive RING 1; LOG2—loss of glutamine dumper 2; HSP90C—chloroplast heat shock protein 90; RGLG—RING domain ligase; PIN2—pin-formed 2; UBC13—ubiquitin-conjugating enzyme UBC13; RD21—responsive to desiccation 21; *TaDIS1*—*Triticum aestivum* drought-induced SINA protein 1; TaSTP—*Triticum aestivum* salt tolerant protein; CHYR1—CHY zinc-finger and RING protein1; *SARD1*—SAR deficient 1; CBF60g—calmodulin-binding protein 60-like g; SA—salicylic acid; ICS1—isochorismate synthase 1; *PR* genes—pathogenesis-related genes; JA—jasmonic acid; *SNC2*—suppressor of NPR1, constitutive 2; MAPK—mitogen-activated protein kinase. A solid arrow shows a promoting effect or positive regulation; a dotted arrow shows that the specific mechanism of action is unclear; a horizontal line shows inhibition or negative regulation; a double arrow shows interacting proteins.

**Figure 3 ijms-23-02308-f003:**
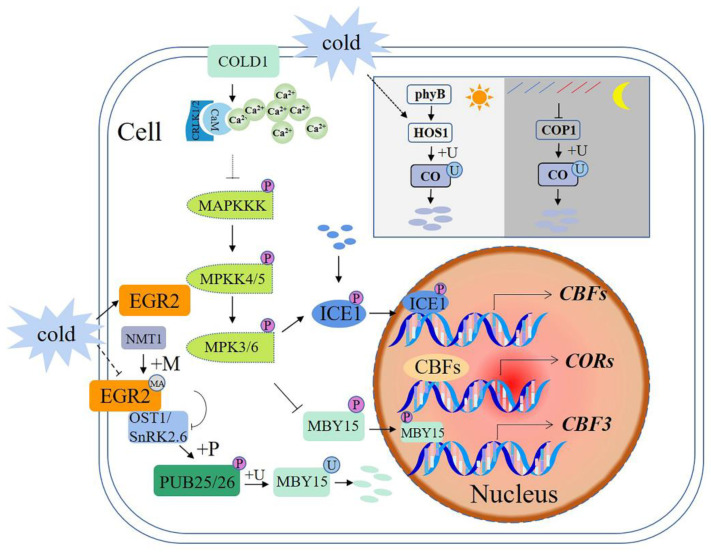
Schematic diagram of E3 ubiquitin ligases in signaling pathways involved in plant responses to cold stress. +P—phosphorylation; +U—ubiquitination; +MA—*N*-myristoylation; *CORs*—cold-regulated genes; *CBF*—*C*-repeat binding factor; COLD1—chilling-tolerance divergence 1; CBLs—calcineurin B-like proteins; CIPKs—CBL-interacting protein kinases; ICE1—*CBF* expression 1; OST1—open stomata1; EGR2:—lade-E growth-regulating 2; NMT1—myristoyltransferase; CO—constans; COP1—constitutive photomorphogenic 1. A solid arrow shows a promoting effect or positive regulation; a dotted arrow shows that the specific mechanism of action is unclear; a horizontal line shows inhibition or negative regulation.

**Figure 4 ijms-23-02308-f004:**
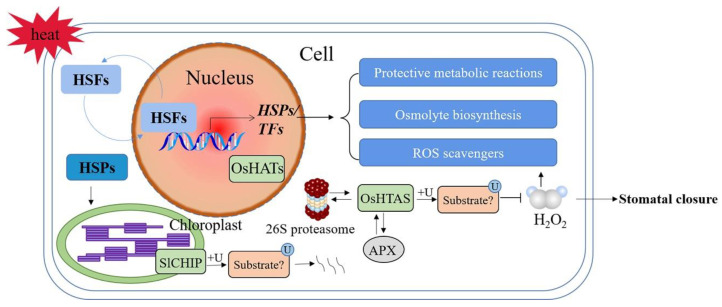
Schematic diagram of E3 ubiquitin ligases in signaling pathways involved in plant responses to heat stress. +U—ubiquitination; ROS—reactive oxygen species; HSFs—heat shock TFs; HSPs—heat shock proteins; *SlCHIP*—*Solanum lycopersicum* carboxyl terminus of the HSC70-interacting proteins; OsHTAS—*Oryza sativa* heat tolerance at seedling stage; H_2_O_2_—hydrogen peroxide. A solid arrow shows a promoting effect or positive regulation; a horizontal line shows inhibition or negative regulation; a double arrow shows interacting proteins.

## Data Availability

Not applicable.

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
