# Peer review of "Roles of E3 Ubiquitin Ligases in Plant Responses to Abiotic Stresses"

_ijms, 2022, doi:10.3390/ijms23042308_

Round 1
Reviewer 1 Report
In this manuscript author reviewed about the roles of E3 ubiquitin-ligases in plant response to abiotic stresses. Ubiquitin ligases are crucial in substrate recognition during this ubiquitin modification process. In this review, the molecular mechanisms of plant response to abiotic stresses elucidated from the perspective of ubiquitin ligases are highlighted to help provide a better understanding of plant response to abiotic stresses. The manuscript is very well balanced and written well. However, It lacks refinements, and there is a scope for improvement. Make sure all the gene names will be in italic fonts.
Chane at L27 from
absisic to Abscisic.
L31 molecules, but to molecules but.
L38 motifs, but to motifs but.
L41 reticulum associated to reticulum-associated.
L49 ubiquintinated proteins to ubiquitinated proteins.
L78 Ubiquintin to Ubiquitin. Change this in all manuscript.
L92 stress response in initiated to stress response is initiated.
L131 environment conditions to environmental conditions.
L139 led to the higher concentration to led to a higher concentration.
L150 calcium binding proteins to calcium-binding proteins.
L153 flowering time regulator GIGANTEA to flowering time regulator, GIGANTEA.
L154 in absence of to in the absence of.
L158 A MAPK cascades to A MAPK cascade.
L164 E3 fuctions to E3 functions.
L164 Remove space from of downstream.
L182 mutants are or mutant is.
L222 located in the to is located in the.
L225 that can binds proteasome to that can binds proteasome.
L226 particle which to particle, which.
L227 an ubiquitination substrate to a ubiquitination substrate.
L263 plants was higher to plants were higher.
L273 seeding germination to seedling germination.
L329 enhances MYB15 degradation to enhance MYB15 degradation.
L332 interaction, but to interaction but.
L338 low temperature to low-temperature.
L341 light, but to light but.
L357 heat responsive to heat-responsive.
L397 the environment stressors to the environmental stressors.
Author Response
In this manuscript author reviewed about the roles of E3 ubiquitin-ligases in plant response to abiotic stresses. Ubiquitin ligases are crucial in substrate recognition during this ubiquitin modification process. In this review, the molecular mechanisms of plant response to abiotic stresses elucidated from the perspective of ubiquitin ligases are highlighted to help provide a better understanding of plant response to abiotic stresses. The manuscript is very well balanced and written well. However, It lacks refinements, and there is a scope for improvement. Make sure all the gene names will be in italic fonts.
Authors’ Response: Thanks for the reviewer’s comments. Some content has been added, so the number of lines has changed and is now indicated where the changes were made. We have checked the gene names and made sure that all gene names are in italic fonts. We have also modified the writing and added L196-L202 and L270-L282 to enhance the coherence and integrity of the manuscript. Best wishes.
Chane at L27 from
Authors’ Response: Sorry for the confusion. we have corrected it in L27.
absisic to Abscisic.
Authors’ Response: Thanks. We have modified absisic to Abscisic in L27.
L31 molecules, but to molecules but.
Authors’ Response: Thanks. We have modified it in L30.
L38 motifs, but to motifs but.
Authors’ Response: Thanks. We have modified it in L37.
L41 reticulum associated to reticulum-associated.
Authors’ Response: Thanks. We have modified it in L41.
L49 ubiquintinated proteins to ubiquitinated proteins.
Authors’ Response: Thanks. We have modified it in L49.
L78 Ubiquintin to Ubiquitin. Change this in all manuscript.
Authors’ Response: Thanks. We have made the change in L78 and throughout the manuscript.
L92 stress response in initiated to stress response is initiated.
Authors’ Response: Thanks. We have modified it in L93.
L131 environment conditions to environmental conditions.
Authors’ Response: Sorry for the confusion. we have corrected it in L132.
L139 led to the higher concentration to led to a higher concentration.
Authors’ Response: Thanks. We have modified the to a in L140.
L150 calcium binding proteins to calcium-binding proteins.
Authors’ Response: Thanks. We have modified it in L151.
L153 flowering time regulator GIGANTEA to flowering time regulator, GIGANTEA.
Authors’ Response: Thanks. We have modified it in L155.
L154 in absence of to in the absence of in L155.
Authors’ Response: Thanks. We have modified it in 156.
L158 A MAPK cascades to A MAPK cascade in L160.
Authors’ Response: Thanks. We have modified it in L160.
L164 E3 fuctions to E3 functions.
Authors’ Response: Thanks. We have modified it in L167.
L164 Remove space from of downstream.
Authors’ Response: Thanks. We have deleted it in L167.
L182 mutants are or mutant is.
Authors’ Response: Thanks. We have modified it in 180.
L222 located in the to is located in the.
Authors’ Response: Thanks, but here we cannot add another verb because the verb for this sentence is ubiquitinates. So we have revised the writing to make this sentence clear (please see L234).
L225 that can binds proteasome to that can bind proteasome.
Authors’ Response: Thanks. We have modified binds to bind in L237.
L226 particle which to particle, which.
Authors’ Response: Thanks. We have modified it in L237.
L227 an ubiquitination substrate to a ubiquitination substrate.
Authors’ Response: Thanks. We have modified an to a in L239.
L263 plants was higher to plants were higher.
Authors’ Response: Thanks. We have modified was to were in L305.
L273 seeding germination to seed germination.
Authors’ Response: Thanks. We have modified seeding to seed in L314.
L329 enhances MYB15 degradation to enhance MYB15 degradation.
Authors’ Response: Thanks, but here phosphorylation is singular, so it should be enhances (L373).
L332 interaction, but to interaction but.
Authors’ Response: Thanks. We have modified it in L377.
L338 low temperature to low-temperature.
Authors’ Response: Thanks. We have modified it in L382.
L341 light, but to light but.
Authors’ Response: Thanks. We have modified it in L387.
L357 heat responsive to heat-responsive.
Authors’ Response: Thanks. We have modified it in L403.
L397 the environment stressors to the environmental stressors.
Authors’ Response: Thanks. We have modified it in L447.
Thank you again for your advice
Reviewer 2 Report
Dear Authors,
the present review is very rich in aspects related to the action of E3 ubiquitin-ligases in the response to different stress conditions in plants. The number of recent articles cited also denotes very thorough work.
My only reviews concern:
- keywords: Abscisic to write correctly
- Figure 1: graphically it goes very well but the role related to sal stress is not well defined, while drought stress is highlighted, which is however the subject of the next paragraph. Please check the graphics of this figure well. If you want to insert the two stresses in the same figure, try to make it more readable.
Author Response
My only reviews concern:
- keywords: Abscisic to write correctly
Authors’ Response: Thank you. We have modified absisic to Abscisic in L27.
Figure 1: graphically it goes very well but the role related to sal stress is not well defined, while drought stress is highlighted, which is however the subject of the next paragraph. Please check the graphics of this figure well. If you want to insert the two stresses in the same figure, try to make it more readable.
Authors’ Response: Thank you for your suggestion. We have modified Figure1 to show salt stress and drought stress in Figure 1. As you can see, Figure 1 is focused on salt stress, which crosstalks with salt stress. We have also improved the legend to make this clear. Figure 2 is focused on drought stress. We do not intend to merge the two figures into one. In order to make the two parts have better continuity, we added L196-L202 and L270-L282 Best wishes. Thank you again for your advice.
Reviewer 3 Report
- The whole manuscript must be checked by a Native English speaker. since there are many mistakes, concerning the use of the English language.
- The abstract must be rewritten since there are a lot of mistakes. Even the first words do not make any sense. Lines 24, 25, 26 must be checked again and be more understandable since they do make any sense.
- Line 86. It is Post-Translational Modifications (PTMs) not mechanisms. Modifications!
- Line 111, mammals, not manmanls.
- Lines, 142-145, must be re-written since the given lines are hard to read and confuse the reader.
- Line 149, add word derived. "A calcium-derived signal activates...."
- Lines, 154-155. Rewrite sentence, the phrase "in check" is not appropriate.
- Lines 157-158 must be checked and be rewritten.
- In all Figures, check the spaces (gaps) between the words. In many, there are double spaces.
- Line 166 (Figure 1), "dotted arrow if the specific mechanism of action is unclear"
- Line 179. What is this question?? This must be converted into a sub-paragraph, 3.1.1 with a title.
- Line 203- 204, their place is to the conclusion part of the manuscript.
- Lines 205 - 229 are irrelevant with how the SOS and MAPK pathways relate to protein ubiquitination. This section must be within the previous section 3.1 salt stress.
- Line 232, check the double word space
- line 240, rephrase the line, "To activate" must be altered with another phrase.
- Lines 285 - 297 are not relevant to drought stress. They provide data related to biotic stress (pathogens).
- Line 299, rephrase.
- Line 308 (Figure 2) "in plant response to cold stress"
- Line 310 (Figure 2) dotted arrow if the specific mechanism of action is unclear"
- Line 316, rephrase "Cold stress is sensed via ...."
- Line 324, change "tolerance" with response
- Line 325. This question is not appropriate. Convert is to a title and make it a subparagraph (3.3.1)
- Line 338, check double space between words
- Line 339 - 349, This paragraph is not related to negative regulators. Add it to the previous section 3.3.
- Line 366. Rephrase these lines
- Line 370. Check the line "...proteins that are generated during heat stress. Choose another word for generated
- line 373, double word space
- Line 383. No data at all concerning the interaction of ubiquitin-proteasome pathway and heat shock proteins. Please check this reference, https://doi.org/10.1111/pdi.12120. Provide more data concerning this topic.
- Lines, 394-396. Rephrase.
The strong points of the current manuscript are the figures, that provide an overview of the cascade of events that take place under each abiotic stress syndrome.
Author Response
Comments and Suggestions for Authors
The whole manuscript must be checked by a Native English speaker. since there are many mistakes, concerning the use of the English language.
Authors’ Response: Thanks for the reviewer’s comments. The entire manuscript has been proof-read by a native English speaker. Some content has been added, so the number of lines has changed and is now indicated where the changes were made. Best wishes.
The abstract must be rewritten since there are a lot of mistakes. Even the first words do not make any sense. Lines 24, 25, 26 must be checked again and be more understandable since they do make any sense.
Authors’ Response: Thanks. We have revised the summary in L18, L25.
Line 86. It is Post-Translational Modifications (PTMs) not mechanisms. Modifications!
Authors’ Response: Thanks. We have modified this sentence in L86 and L87.
Line 111, mammals, not manmanls.
Authors’ Response: Thanks. We have modified it in L113.
Lines, 142-145, must be re-written since the given lines are hard to read and confuse the reader.
Authors’ Response: Thanks. We have rewritten this sentence in L143-145.
Line 149, add word derived. "A calcium-derived signal activates...."
Authors’ Response: Thanks. We have added “derived” in L151.
Lines, 154-155. Rewrite sentence, the phrase "in check" is not appropriate.
Authors’ Response: Thanks. We have motified in L157.
Lines 157-158 must be checked and be rewritten.
Authors’ Response: Thanks. We have rewritten this sentence in L159 and L160.
In all Figures, check the spaces (gaps) between the words. In many, there are double spaces.
Authors’ Response: Thanks. We have checked and deleted the blank space in all figures.
Line 166 (Figure 1), "dotted arrowif the specific mechanism of action is unclear"
Authors’ Response: Thanks. We have modified it in L169.
Line 179. What is this question?? This must be converted into a sub-paragraph, 3.1.1 with a title.
Authors’ Response: Thanks. This question is intended as a connecting sentence, but it may not be clear. Now we've changed and added a title in L147 and L196..
Line 203- 204, their place is to the conclusion part of the manuscript.
Authors’ Response: Thanks. We have added this sentence in L455-456.
Lines 205 - 229 are irrelevant with how the SOS and MAPK pathways relate to protein ubiquitination. This section must be within the previous section 3.1 salt stress.
Authors’ Response: Thanks. We have added tertiary titles to distinguish it from the SOS and pathways in L146, 193, 200, 209, 226.
Line 232, check the double word space
Authors’ Response: Thanks. We have checked and deleted the blank space.
line 240, rephrase the line, "To activate" must be altered with another phrase.
Authors’ Response: Thanks. We have changed “activate” with “initiate” in L252.
Lines 285 - 297 are not relevant to drought stress. They provide data related to biotic stress (pathogens).
Authors’ Response: Thank you. This part of WRKY70 is really about biotic stress. We wrote this because WRKY70 is involved in drought stress and WRKY70 is also a defense-related transcription factor. Downstream genes have been reported only through biotic stress studies, so in order to complete this example, biotic stress is supplemented. We have added a sentence “WRKY70 is known to be involved in plant response to osmotic stress and it is also a defense-related transcription factor”.
Line 299, rephrase.
Authors’ Response: Thanks. We have rewritten this sentence in L344.
Line 308 (Figure 2) "in plant response tocold stress"
Authors’ Response: Thanks. We have added the word “to” in L353.
Line 310 (Figure 2) dotted arrowif the specific mechanism of action is unclear"
Authors’ Response: Thanks. We have modified it in L354.
Line 316, rephrase "Cold stress is sensed via...."
Authors’ Response: Thanks. We have modified it in L361.
Line 324, change "tolerance" with response
Authors’ Response: Thanks. We have changed “tolerance” with “response” in L375.
Line 325. This question is not appropriate. Convert is to a title and make it a subparagraph (3.3.1)
Authors’ Response: Thanks. We have deleted this sentence and added tertiary titles in L360, L376, L384.
Line 338, check double space between words
Authors’ Response: Thanks. We have deleted the blank space.
Line 339 - 349, This paragraph is not related to negative regulators. Add it to the previous section 3.3.
Authors’ Response: Thanks. This paragraph is an example of a ubiquitin ligase associated with flowering, and we have added the title in L384.
Line 366. Rephrase these lines
Authors’ Response: Thanks. We have rewritten these lines in L411-414.
Line 370. Check the line "...proteins that are generated during heat stress. Choose another word for generated
Authors’ Response: Thanks. We have checked and modified generated to produced in L418.
line 373, double word space
Authors’ Response: Thanks. We have deleted the blank space.
Line 383.No data at all concerning the interaction of ubiquitin-proteasome pathway and heat shock proteins. Please check this reference, https://doi.org/10.1111/pdi.12120. Provide more data concerning this topic.
Authors’ Response: We are very sorry that we did not find the reference you provided, but we have now added this reference [128] in L430 and L431.
Lines, 394-396. Rephrase.
Authors’ Response: Authors’ Response: Thanks. We have rewritten this sentence in L444 and L445. Thank you again for your advice.
Round 2
Reviewer 1 Report
I am happy with the reviewer's comments. The manuscript looks refined now and can be accepted in its current format.
Reviewer 3 Report
The authors have performed all the necessary changes to the manuscript. It is worthy of being published in its present form.